# Impact Response of Re-Entrant Hierarchical Honeycomb

**DOI:** 10.3390/ma16227121

**Published:** 2023-11-10

**Authors:** Jinming Lian, Zhenqing Wang

**Affiliations:** College of Aerospace and Civil Engineering, Harbin Engineering University, Harbin 150001, China; 1094153544@hrbeu.edu.cn

**Keywords:** re-entrant honeycombs, negative Poisson’s ratio, energy absorption capacity, dynamic crushing, in-plane impact, out-of-plane impact

## Abstract

Here, hexagonal and triangular lattices are layered and merged into a re-entrant honeycomb to replace each cell wall of the re-entrant honeycomb. In order to study the crushing behavior of the new variable-angle-variable-substructure-number-gradient honeycomb, a finite element analysis of in-plane and out-of-plane crushing was carried out. The effects of different gradient parameters on the deformation mode and extrusion response were discussed, respectively. The results show that different grading parameters have different effects on the crushing behavior of honeycombs for in-plane and out-of-plane crushing. Compared with out-of-plane crushing, the influence of the hierarchical structure on the in-plane crushing deformation mode and the increase in platform stress are much larger. Compared with the ordinary honeycombs, changing the substructure angle does not necessarily improve the platform stress of the honeycomb. From the perspective of platform stress, the layered structure has different effects on the improvement of honeycomb energy absorption; the maximum platform stress of the honeycomb is increased.

## 1. Introduction

The latest developments in the aerospace, electronics, automotive and naval industries have led to an extraordinary demand for lightweight materials with excellent mechanical properties [1]. Promising candidates include various forms of honeycomb structures, which are originally derived from natural honeycomb structures [2] and have excellent properties in terms of specific stiffness, strength, impact resistance and energy absorption [3,4]. Honeycomb, as one of the most common cell structures, has been widely studied [1]. For example, metal honeycombs have been widely used in the field of impact resistance and energy absorption through internal plastic deformation. Various honeycomb structures, including hexagons, circles, squares and triangles, have been experimentally, theoretically and numerically explored [5,6,7,8]. In order to improve the crashworthiness and energy absorption capacity of the honeycomb, several optimization strategies are introduced and studied [4,9]. Therefore, honeycombs with novel configurations are constructed and explored, such as hierarchical honeycombs [10] and multicellular honeycombs [11]. These improvements enhance or change the mechanical properties of the honeycomb in various ways.

For the current honeycomb structure, ordinary honeycomb is most widely used in various industrial fields. Hu, Lu et al., 2022 [12] found that the addition of STF in honeycomb cells can significantly improve the energy absorption of honeycomb filled with STF, thus effectively preventing the premature collapse of honeycomb cell walls. Li, Deng et al., 2013 [13] and Hedayati, Sadighi et al., 2016 [14] developed a new type of honeycomb, which has better mechanical properties than traditional honeycomb. Peng, Marzocca et al., 2023 [15] established a numerical model for the mechanical properties of G-Honeycomb and P-Honeycomb lattices. The results show that the in-plane elastic modulus of the two structures is higher than that of the traditional square honeycomb structure. Duan, Tao et al., 2018 [16] used a geometric parameter to describe square and hexagonal honeycombs with variable thickness elements. Compared with traditional honeycombs, honeycombs with variable-thickness honeycomb edges have better compressive mechanical properties. In terms of bionic honeycombs, Zhang, Yu et al., 2022 [17] and Deng, Qin et al., 2022 [18] designed a new type of honeycomb structure inspired by the beetle wing sheath and woodpecker beak, respectively. Compared with traditional honeycombs, new bionic honeycombs have excellent energy absorption effects.

With the development of advanced technology, it has been difficult for the ordinary honeycomb to meet the needs of industry, and the demand for hierarchical honeycomb development is increasing. Through the combination of various honeycombs, Lin, Zhang et al., 2015 [19] and Zhang, Weng et al., 2022 [20] found that the proper combination of honeycombs can improve the mechanical properties of honeycomb structures. Qiao and Chen 2016 [21] proposed a two-scale method to obtain the analytical expression of the quasi-static collapse stress of the gradient honeycomb in two directions. Combined with the theory of momentum conservation, the analysis of the quasi-static collapse stress model is extended to dynamic crushing. By replacing the vertices of the honeycomb structure, Zhang, Fei et al., 2020 [22] and Wang, Li et al., 2019 [23] improved the energy absorption capacity of the honeycomb. Lu, Tan et al., 2020 [24] proposed a concept of “effective Poisson ‘s ratio”, deduced the analytical expression of “effective Poisson ‘s ratio “, and described the relationship between the geometric size of a four-handed honeycomb and its intergrowth. Li, Lu et al., 2020 [25] studied the dynamic mechanical and energy absorption responses of three samples at two stacking angles of a single-layer hexagonal aluminum honeycomb and a combined hexagonal aluminum honeycomb, and proposed an empirical formula of uniform size to describe the effects of honeycomb density and strain rate on plateau stress. Liu, Zhang et al., 2022 [26] and Fang, Sun et al., 2018 [27] obtained new honeycombs with higher plateau stress than traditional honeycombs by replacing the cell wall of traditional honeycombs with hexagons and triangles.

In recent years, with the introduction of the concept of metamaterials, negative Poisson‘s ratio honeycomb, as a mechanical metamaterial, has attracted more and more attention due to its unique mechanical properties and good energy absorption capacity. Li, Lu et al., 2018 [28] established a theoretical model for the mechanical response of honeycomb structures considering the coupling effect of shear stress and axial stress. Qu, Wang et al., 2021 [29] deduced the volume, in-plane elastic modulus and unit cell area of the double v-wing honeycomb, which became part of the theoretical basis of the new equivalent method. Tan, He et al., 2019 [30] combined the characteristics of a concave–convex structure and hierarchical honeycomb, and proposed two concave-graded honeycombs by replacing the cell wall of the concave honeycomb with a regular hexagon substructure (RHH) and an equilateral triangle substructure (RHT). Wan, Ohtaki et al., 2004 [31] proposed a theoretical method for predicting the negative Poisson’s ratio based on the large deflection model. The deflection curve, strain and Poisson’s ratio equations of the inclined member of the re-entrant element in two orthogonal directions are derived. The deformation shape of the inclined member of the re-entrant element is calculated. Under large deformation conditions, the negative Poisson’s ratio is no longer a constant. It will change significantly with the different cells. Tatlier 2021 [32] found that the re-entrant honeycomb structure with 90 degrees alignment exhibits a better energy absorbing potential than other re-entrant honeycomb structures. Mustahsan, Khan et al., 2022 [33] propose an improved reentry honeycomb structure with a higher negative Poisson’s ratio. The hybrid design strategy proposed by Usta, Zhang et al., 2023 [34] can be used to manipulate the crushing mechanism and improve the crushing performance. Agrawal, Joo et al., 2022 [35] made polyurethane aerogels into re-entrant honeycombs to obtain higher elasticity and flexibility than the corresponding aerogel monomers. The results of Montazeri, Bahmanpour et al., 2023 [36] show that the foam-filled TPU-based aerator has good performance. Öztürk, Baran et al., 2022 [37] improved the re-entrant cells, and the modified cells were improved in stiffness, energy absorption capacity and plasticity. The re-entry model established by Usta, Türkmen et al., 2022 [38] shows that the specific energy absorption (SEA) is increased compared with the foam core. Tajalsir, Mustapha et al., 2022 [39] revealed the interaction between the key geometric features and impact properties of HHRH structures at low, medium and high crushing speeds through explicit finite element analysis. Menon, Dutta et al., 2022 [40] propose two novel design concepts of Oriented Re-entrant Structures (ORS) and Assorted Re-entrant Structures (ARS) for an improved load-bearing response. Chikkanna, Krishnapillai et al., 2023 [41] discussed the geometric-dependent printing changes associated with 3D printing to help users create re-entrant diamond-assisted metamaterials.

In this paper, the in-plane and out-of-plane crashworthiness and energy absorption effects of re-entrant honeycombs with triangular substructures and hexagonal substructures are studied respectively, and the effects of the number of substructures and honeycomb angles on the energy absorption effect of honeycombs are studied. In Section 2, the arrangement and configuration of hierarchical honeycombs are introduced. In Section 3, the finite element model is established and verified. In Section 4, the numerical results of the finite element simulation are given, and the deformation mode, extrusion response and energy absorption effect of a gradient honeycomb are analyzed and discussed, respectively. In Section 5, we summarize the results and give some meaningful conclusions.

## 2. Hierarchical Honeycomb

In this section, the structural description and relative density distribution of hierarchical honeycombs will be discussed. On this basis, the crashworthiness parameters reflecting the extrusion performance are described in detail, and the finite element model of the hierarchical honeycomb is established.

### 2.1. Description of Honeycomb Structure

Based on the concept of a layered design of materials, the hexagonal and triangular substructures are used to replace the cell wall of the re-entrant honeycomb to construct a hierarchical honeycomb structure. For honeycombs that change the number of substructures, the substructures cited are regular hexagons and regular triangles. Four types of hierarchical honeycombs are considered, as shown in Figure 1. Two honeycombs are composed of triangular substructures, and the other two are composed of hexagonal substructures. In addition, uniform hierarchical honeycombs with triangular substructures and hexagonal substructures are compared. As shown in Figure 2, the dimensions L, H and b are the horizontal length, vertical length and out-of-plane depth of the uniformly hierarchical honeycomb, respectively. By changing the number of subunits contained in different honeycomb structures, a hierarchical honeycomb is established and the hierarchical honeycomb structure is obtained by two geometric parameters. The first parameter $N_{i}$ represents the number of substructures on the horizontal edge (as shown in Figure 2), where *i* represents the level of honeycomb. The second parameter $M_{i}$ is the number of substructures on the inclined edge.

The number of honeycomb subunits of the first-level hexagonal substructure is set to $N_{HEX-1}$ = 8, $M_{HEX-1}$ = 4. In addition, the number of hexagonal honeycomb subunits is fixed at $N_{HEX-2}$ = 20, $M_{HEX-2}$ = 10 and $N_{HEX-3}$ = 32, $M_{HEX-3}$ = 16. In addition, the geometric parameters of the TRI honeycomb are set as $N_{TRI-1}$ = 12, $M_{TRI-2}$ = 6, $N_{TRI-2}$ = 24, $M_{TRI-2}$ = 12 and $N_{TRI-3}$ = 48, $M_{TRI-3}$ = 24, corresponding to levels 1, 2 and 3, respectively. *θ* is the angle of substructure and *l* is the side length of substructure as shown in Figure 3. $l_{HEX-i}$ and $l_{TRI-i}$ satisfy the following:(1)$$l_{HEX-i}N_{HEX-i}=l_{TRI-i}N_{TRI-i}$$

Here, for all regular hexagon and regular triangle substructure hierarchical honeycombs, the length $l_{0}$ and the edge thickness C of each unit are set to be constants.

For the honeycombs with different substructure angles, the angle of the first-level honeycomb substructure is changed, and each structure considers four kinds of hierarchical honeycombs. In addition, the uniform hierarchical honeycombs with regular triangle and regular hexagon substructures are compared. As shown in Figure 4, for HEX honeycombs, the substructure angles are 100°, 110°, 130° and 140°. In addition, the substructure angles of TRI honeycombs are 50°, 55°, 65° and 140°. For the variable-angle honeycomb, it also satisfies Equation (1).

### 2.2. Relative Density Distribution

Similar to the typical honeycomb material, an important structural parameter that determines the mass distribution of the gradient honeycomb is the relative density of the honeycomb. It can be approximated as the ratio of the density $\rho_{i}$ of the *i*-level hierarchical honeycomb to the density $\rho_{m}$ of the base material (the material that makes the honeycomb). Additionally, the relative density can also be calculated by the volume of the honeycomb over the volume occupied by the unit cell, i.e.,
(2)$${\bar{\rho }}_{i}=\frac{\rho_{i}}{\rho_{m}}=\frac{Vi}{V_{m}}$$

In the formula, $V_{i}$ is the honeycomb volume of the *i*-th level and $V_{m}$ is the honeycomb unit volume. In this study, for all hierarchical honeycombs, the volume of the cell is the same, which can be expressed as:(3)$$\left\{\begin{array}{c} V_{HEX}=4M_{i}l_{i}\sin\theta \left(2N_{i}-M_{i}\right)\left(1-\cos\theta \right)\\ V_{TRI}=2M_{i}l_{i}\sin\theta \left[2N_{i}-M_{i}\right] \end{array}\right.$$

In the formula, Mi and Ni are the number of inclined edge substructures and the number of horizontal edge substructures of the *i*-level honeycomb, respectively, and θ and $l_{i}$ are the angle and side length of substructures, respectively, as shown in Figure 4.

For different substructures of the hierarchical honeycomb configuration, the volumes are different. The honeycomb volume of different substructures can be expressed as:(4)$$\left\{\begin{array}{c} V_{HEX}=(12N_{i}i+24M_{i}i-4M_{i}-2N_{i}+10i-30i^{2})l_{i}h_{i}\\ V_{TRI}=[2\cos\theta (2^{i+1}N+2^{i+2}M+2N-3\times 2^{i}-5\times 2^{2i-1})+(2^{i+2}N+2^{i+3}M+4M-2^{i+1}-5\times 2^{i})]l_{i}h_{i} \end{array}\right.$$

Here, $h_{i}$ represents the wall thickness. By introducing Equations (3) and (4) into Equation (2), the relative density formula of the layered honeycomb can be expressed as:(5)$$\left\{\begin{array}{c} {\bar{\rho }}_{HEX}=\frac{12N_{i}i+24M_{i}i-4M_{i}-2N_{i}+10i-30i^{2}}{4M_{i}l_{i}\sin\theta \left(2N_{i}-M_{i}\right)\left(1-\cos\theta \right)}h_{i}\\ {\bar{\rho }}_{TRI}=\frac{2\cos\theta (2^{i+1}N+2^{i+2}M+2N-3\times 2^{i}-5\times 2^{2i-1})+(2^{i+2}N+2^{i+3}M+4M-2^{i+1}-5\times 2^{i})}{2M_{i}l_{i}\sin\theta \left[2N_{i}-M_{i}\right]}h_{i} \end{array}\right.$$

### 2.3. Relative Density Distribution

In order to characterize the crashworthiness of the gradient honeycomb under compression, several different crashworthiness criteria can be compared. Firstly, once a typical stress–strain curve is obtained, the total energy absorption EA can be calculated.
(6)$$EA=V_{t}\int_{0}^{\varepsilon_{d}}\sigma \left(\varepsilon \right)d\varepsilon$$
where $V_{t}$ is the volume occupied by the entire hierarchical honeycomb structures, σ and ε are the crushing stress and strain of the hierarchical honeycomb under dynamic load, respectively, and $\varepsilon_{d}$ is the densification strain.

In the finite element simulation process, the crushing stress *σ* and strain *ε* can be obtained by the following:(7)$$\sigma =\frac{F}{Lb}$$
(8)$$\varepsilon =\frac{\Delta H}{H}$$

Here, *F* is the reaction force of the impact end and Δ*H* is the moving distance of the impact end in the vertical direction.

The dynamic platform stress $\sigma_{P}$ in the platform stage is another key parameter for evaluating the crashworthiness of the honeycomb. It can be given as follows:(9)$$\sigma_{p}=\frac{1}{\varepsilon_{d}-\varepsilon_{0}}\int_{\varepsilon_{0}}^{\varepsilon_{d}}\sigma \left(\varepsilon \right)d\varepsilon$$

In the formula $\varepsilon_{0}$ is the initial strain when the stress reaches the platform. The larger the $\sigma_{p}$ value, the better the honeycomb energy absorption effect.

## 3. Numerical Analysis

In order to study the extrusion behavior of gradient honeycomb, numerical analysis was carried out. The crushing process of gradient honeycombs with different geometric parameters was simulated by the explicit finite element program ABAQUS/explicit. In this paper, the in-plane crushing and out-of-plane crushing simulation of the honeycomb are carried out, respectively. The honeycomb is composed of a large number of micro units, which makes it always able to meet the simulation calculation challenges based on all the details considered in the complete model.

### 3.1. In-Plane Crushing Finite Element Model

As shown in Figure 5, the in-plane crushing finite element model consists of two rigid walls and a honeycomb sandwiched between the rigid walls. During the extrusion process, the top rigid wall moves towards the honeycomb at a constant speed of 100 mm/s, while the bottom rigid wall is completely fixed. The honeycomb is composed of 4 units in the x direction and 6 units in the y direction. The total horizontal length L = 997.7 mm, the total vertical length H = 888 mm, and out-of-plane depth b = 2 mm. For the honeycomb with a variable-angle hexagonal substructure, the corresponding L is 1134.5 mm, 1082.53 mm, 882.48 mm and 740.49 mm, and H is 692.19 mm, 793.21 mm, 973.67 mm and 1047.63 mm, respectively. In addition, the variable-angle triangular substructure honeycomb L and H are the same as above. In order to prevent the out-of-plane bending of the honeycomb, the displacement of the honeycomb model in the z direction is fixed. The honeycomb is meshed by the hourglass-controlled four-node simplified integrated shell element S4R, which has five integration points on the entire wall thickness. The contact between the rigid wall and the honeycomb is simulated by surface-to-surface contact. A general contact algorithm is defined to simulate the self-contact of the unit. Friction is defined as smooth friction. In fact, all the materials used in this study are ideal plastic materials. The density of the aluminum alloy material used in this paper is $\rho =2700\;{\mathrm{kg}/m}^{3}$, Young’s modulus E = 70 Gpa, Yield stress $\sigma_{ys}$ = 110 MPa and Poisson’s ratio ν=0.3.

### 3.2. Finite Element Model of Out-of-Plane Crushing

As shown in Figure 6, compared with conventional honeycombs, hierarchical honeycombs become more complex. If a detailed model is used, it will encounter great computational challenges in the simulation process. Therefore, the unit cell model is used for out-of-plane crushing. As shown in Figure 7, this paper selects the part framed by the red line in the figure and replaces the overall compression with the compression of that part.

### 3.3. Verification of Finite Element Model

Firstly, the convergence test of the element size is carried out. Figure 8 shows the influence of element size on the energy absorption effect of the basic regular honeycomb broken along the y direction. It can be seen that when the element size is reduced to 1 mm, the energy absorption effect gradually stabilizes and converges. At the same time, the calculation time increases exponentially with the decrease in mesh size. Therefore, a 1 mm mesh size is conservatively selected for computational efficiency and result accuracy.

#### 3.3.1. Verification of In-Plane Crushing

In order to verify our in-plane impact simulation model, a benchmark test between the literature [30] and the simulation was performed. In Reference [30], the wall thickness of the hexagonal re-entrant honeycomb (RHH) substructure is 0.19287 mm. The structure is compressed by a rigid plane with a constant speed of 1 m/min in the vertical direction. The platform stress of the honeycomb is shown in Equation (10). Here, $\sigma_{HEX}$ is the collapse stress of the hexagonal substructure. As shown in Figure 9 and Figure 10, and the deformation process and stress of the finite element simulation are compared with the literature results. It should be noted that the deformation and stress results are very consistent with the literature results. Therefore, it has been clearly verified that the finite element model can predict the crushing performance of these graded honeycombs under in-plane impact.
(10)$$\sigma_{in-of-plant}=\frac{4(2\pi +3\sqrt{3}M)}{9\sqrt{3}M(2N-M)}\sigma_{HEX}$$

#### 3.3.2. Validation of Out-of-Plane Crushing

Benchmark tests were performed between the literature [27] and the simulations. In reference [27], the simulation model is loaded by the downward displacement specified at the top surface. Special attention is paid to the allocation of a “smooth step” time function to minimize the inertial effect, so as to simulate the “quasi-static” breaking rate of the experiment. In addition, the simulation is carried out by loading a rigid plate at 0.1 m/s on a vertical structure. The platform stress of the honeycomb is shown in Equation (11). $\sigma_{f}$ is the flow stress of the material and t is the cell wall thickness. B is half of the side length of the honeycomb substructure. As shown in Figure 11 and Figure 12, it can be seen that the deformation and stress results are very consistent with the experimental results. Therefore, it has been clearly verified that the finite element model can predict the crushing performance of these graded honeycombs under out-of-plane impact.
(11)$$\sigma_{out-of-plant}=182.312\sigma_{f}B^{0.5}t^{1.5}$$

## 4. Results and Discussion

The extrusion process along the *y*-axis and *z*-axis directions was simulated and explored, respectively. Here, we consider the honeycombs from the first-level honeycomb to the third-level honeycomb, the hexagonal substructure angle from 100° to 140° and the triangular substructure angle from 50° to 70°. The configuration of all levels of the honeycomb and base honeycomb is shown in Table 1 and Table 2.

### 4.1. In-Plane Crushing of Hierarchical Honeycombs

Figure 13 and Figure 14, respectively, show the crushing behavior of the regular hexagon substructure and regular triangle substructure along the *y*-axis direction.

As shown in Figure 13a and Figure 14a, the local bands are initially generated at the top and bottom of the hexagonal and triangular honeycombs. For the breakage along the *y*-axis direction, the local area is “v” shaped. With the increase in compression, the initial local bands are stacked layer-by-layer until densification. As shown in Figure 13b and Figure 14b, the deformation mode of the second-level hierarchical honeycomb is similar to the overall deformation mode of the regular honeycomb. An obvious difference is that the local band at the bottom is slightly less obvious in the crushing direction, which indicates that the local band is not compressed completely compared with the regular honeycomb. The main reason is that compared with the basic re-entrant cells, the substructure introduced by the layer is difficult to crush and can absorb more energy, so most of the initial impact is absorbed by the upper honeycomb. Therefore, the crushing process of the hierarchical honeycomb can be divided into two parts: (1) in the initial compression stage, the honeycomb is broken, and the substructure has no obvious crushing phenomenon; (2) a further compression stage, crushing the lower structure until densification until densified. As shown in Figure 13c and Figure 14c, as the number of substructures increases, the structural densification process accelerates. This is because the multi-layer substructure has stronger impact resistance, and the boundary of the substages of the two impact processes is gradually cleared. Compared with the hexagonal substructure honeycomb, the bottom deformation zone of the triangular substructure honeycomb is less obvious in the crushing direction with the increase in the number of substructures. This is because the triangular structure has better impact resistance than the six-deformed structure.

The deformation process of the honeycomb when the substructure angle is changed is shown in Figure 15 and Figure 16. It can be seen that with the change in the substructure angle, the deformation mode of the honeycomb also changes. As shown in Figure 15a,b and Figure 16a,b, with the increase in the substructure angle, the local area formed by honeycomb fragmentation is more obvious. This is due to the change in the angle, which makes the inclined edge more prone to rotation deformation when it is impacted. As the angle of the substructure continues to increase, as shown in Figure 15c,d and Figure 16c,d, the honeycomb cell wall is more prone to compression deformation, and the “v”-shaped band formed by the compression of the honeycomb gradually develops to the “x” shape.

#### Crushing Reaction

Figure 17 shows the stress–strain curves of hexagonal and triangular substructures in the plane.

As shown in Figure 17, the stress–strain curve of the in-plane impact contains three different states: the initial linear elastic state, the middle long platform state and the final compression dense state. This is a common trend that has been widely reported and discussed in the literature. It can be seen that compared with the basic re-entrant honeycomb, the hierarchical honeycomb has a higher platform stress. As shown in Figure 17a,b, for the honeycomb, there is a second plateau period when the impact continues. This is because the first plateau period of the re-entrant honeycomb is the rotation and compression of the tilted edge, and then the second plateau period is the densification process of the horizontal edge. As the impact continues, the substructures of the upper and lower bottom sides deform and absorb the impact, resulting in a second plateau period that appears. With the increase in the honeycomb number, the stress of the second platform period also increases.

For the honeycomb, as the substructure angle changes, as shown in Figure 18, the densification of the honeycomb structure is basically not affected by the increase in substructure angle. As shown in Figure 19a,b, it can be seen that the platform stress of the hexagonal substructure honeycomb increases with the increase in the angle until the angle is 130°, where the platform stress is the largest. After that, as the angle increases, the platform stress begins to decrease. The platform stress of the triangular substructure honeycomb also increases first and then decreases as the angle increases. When the angle is 60°, the platform stress is the largest. As shown in Figure 19a,b, the platform stress increases with the increase in the number of substructures. Compared with the hexagonal substructure honeycomb, the triangular substructure honeycomb has higher plateau stress, which is because the triangular substructure is more difficult to compress than the hexagonal substructure. At the same time, it can be seen that with the increase in the number of substructures, the platform period of the honeycomb structure is shortened and the densification process is advanced. For the hexagonal substructure honeycomb, the platform stress of the three-level honeycomb is increased by 72.9% compared to the first-level honeycomb, and the second-level honeycomb is increased by 40%. Similarly, the triangular substructure honeycomb is increased by 25.8% and 7%, respectively.

### 4.2. Out-of-Plane Crushing of Hierarchical Honeycombs

As shown in Figure 20, from the results obtained, the honeycomb is broken layer by layer. As the number of honeycomb word structures increases, the honeycomb layer by layer folds more and more densely. Figure 21 shows the layer-by-layer failure of the variable-angle honeycomb. Compared with the hexagonal substructure, the deformation of the triangular substructure is more regular.

#### Crushing Reaction

Figure 22 and Figure 23 show the stress–strain curves and plateau stress of hierarchical honeycombs with different numbers of substructures and different substructure angles. As shown in Figure 22, the trend of the stress–strain curve in all cases is similar, showing three states: a linear state, plateau state and densification state. It can be seen that with the increase in the number of honeycomb substructures, the platform stress of the honeycomb increases, and the fluctuation of the honeycomb gradually slows down. This is because the substructure becomes smaller, and the influence of the substructure breaking on the overall stress gradually decreases. It can be seen from Figure 23 that the crushing force curve shows periodic fluctuations in the form of alternating peaks and troughs, corresponding to the collapse of honeycomb cells layer by layer. It can be seen that the two substructures have the largest platform stress in the standard substructure form. This is because the honeycomb structure has little effect on the stress of the honeycomb platform when the out-of-plane compression is applied. The in-plane deformation mode of the honeycomb is basically not applicable when the out-of-plane deformation is applied. When the out-of-plane deformation is applied, the deformation mode of the honeycomb is the folding compression of the honeycomb wall, and the cell wall of the substructure at the standard angle is the thickest, so the platform stress is the largest. Figure 24 shows the platform stress of the honeycomb. For the hexagonal substructure honeycomb, the platform stress of the three-level honeycomb is 13.5% higher than that of the first-stage honeycomb, and for the second-level honeycomb it is 2% higher than that of the first-stage honeycomb. Similarly, for the triangular substructure honeycomb it is respectively increased by 77.4% and 27.1%. It can be seen that the improvement of the triangular substructure honeycomb is more obvious. This is because the strength of the triangular substructure is higher than that of the hexagonal substructure. As the number of honeycombs increases, the platform stress increases more obviously.

## 5. Concluding Remarks

By replacing the cell wall of the re-entrant honeycomb with hexagonal and triangular substructures with different angles and numbers of substructures, a class of hierarchical honeycombs is constructed. Then, the crashworthiness of the graded honeycomb under in-plane and out-of-plane compression is analyzed. The effects of the number and angle of substructures on the deformation mode and extrusion response are discussed.

For the hierarchical honeycomb under in-plane impact, as the angle increases, the deformation mode of the hexagonal substructure and triangular substructure becomes obvious, and the local fracture band gradually changes from a “v” shape to an “x” shape. The first-level hexagonal substructure has the maximum platform stress at the angle of 130°, and the triangular substructure honeycomb has the maximum platform stress at the angle of 60°. As the number of substructures increases, the platform stress of the third-level hexagonal substructure honeycomb and the second-level hexagonal substructure honeycomb increases by 72.9% and 40% respectively, compared with the first-level honeycomb, and for the triangular substructure honeycomb it is increased to 25.8% and 7%.

The deformation mode of the hierarchical honeycomb under out-of-plane impact is layer-by-layer stacking, and the triangular substructure honeycomb is more densely stacked than the hexagonal substructure honeycomb. When both honeycombs are in standard shape (regular hexagon, regular triangle), the platform stress is the largest. With the increase in the number of substructures, the platform stress of the third-level hexagonal substructure honeycomb and the second-level hexagonal substructure honeycomb is increased by 13.5% and 2%, respectively, compared with the first-level honeycomb, and for the triangular substructure honeycomb it is increased to 77.4% and 27.1%.

Compared with ordinary honeycombs, variable-angle structures do not always improve honeycombs in terms of platform stress. In terms of platform stress, the improvement in the honeycomb energy absorption effect by hierarchical structures is uneven.

In this paper, the overall influence of honeycomb substructure on the in-plane and out-of-plane impact of honeycomb is systematically studied. The effects of the shape, number and angle of the honeycomb substructure are considered respectively. This provides data support for future honeycomb structure design and industrial production.

## Figures and Tables

**Figure 1 materials-16-07121-f001:**
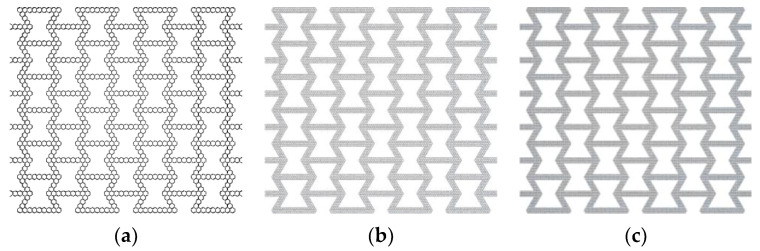
In-plane honeycomb structure model. (**a**) HEX-1; (**b**) HEX-2; (**c**) HEX-3; (**d**) TRI-1; (**e**) TRI-2; (**f**) TRI-3.

**Figure 2 materials-16-07121-f002:**
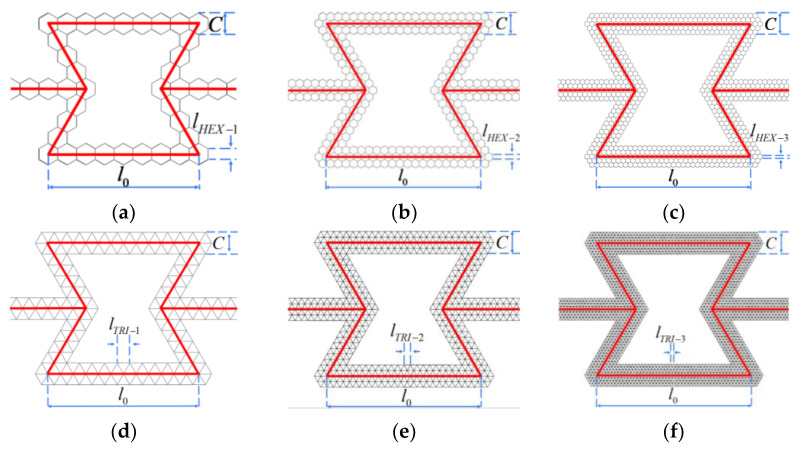
Variable number cell model. (**a**) HEX-1; (**b**) HEX-2; (**c**) HEX-3; (**d**) TRI-1; (**e**) TRI-2; (**f**) TRI-3.

**Figure 3 materials-16-07121-f003:**
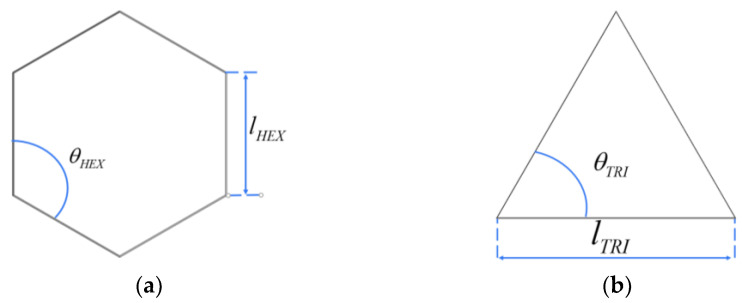
Substructure model. (**a**) Hexagonal substructure; (**b**) Triangular substructure.

**Figure 4 materials-16-07121-f004:**
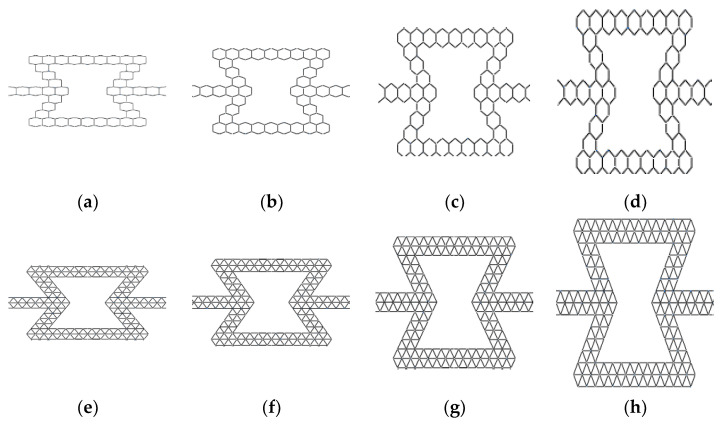
Variable angle cell model. (**a**) HEX-100°; (**b**) HEX-110°; (**c**) HEX-130°; (**d**) HEX-140°; (**e**) TRI-50°; (**f**) TRI-55°; (**g**) TRI-65°; (**h**) TRI-70°.

**Figure 5 materials-16-07121-f005:**
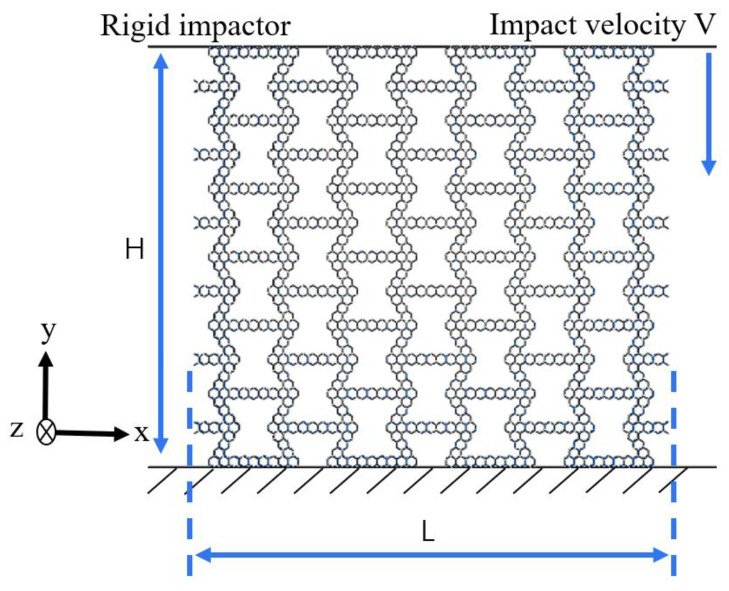
In-plane impact model.

**Figure 6 materials-16-07121-f006:**
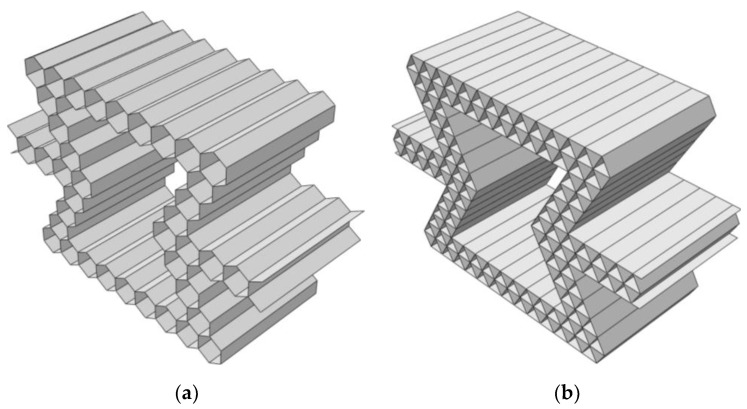
Out-of-plane honeycomb structure model. (**a**) Hexagonal substructure; (**b**) Triangular substructure.

**Figure 7 materials-16-07121-f007:**
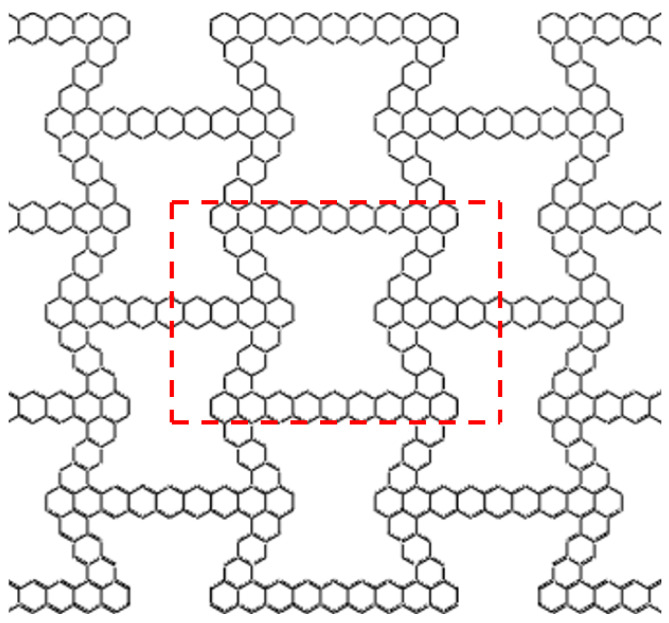
Selection of out-of-plane impact cell.

**Figure 8 materials-16-07121-f008:**
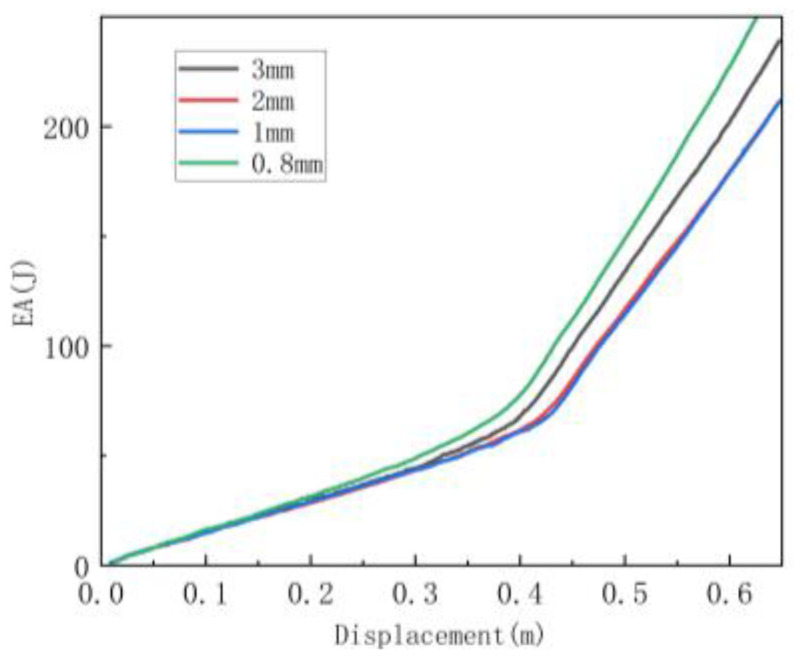
Verification of mesh size.

**Figure 9 materials-16-07121-f009:**
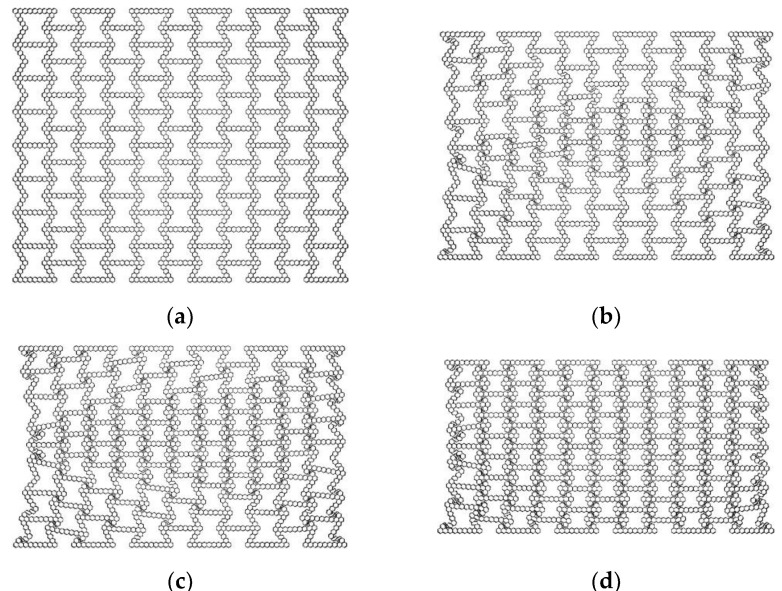
The deformation process of RHH: (**a**) ε=0; (**b**) ε=17; (**c**) ε=26; (**d**) ε=37.

**Figure 10 materials-16-07121-f010:**
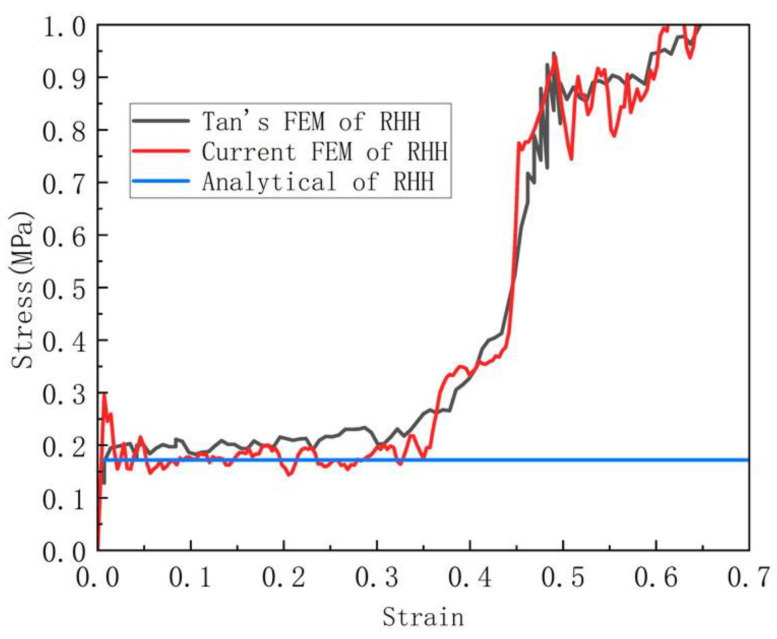
Comparison with Tan’s results.

**Figure 11 materials-16-07121-f011:**
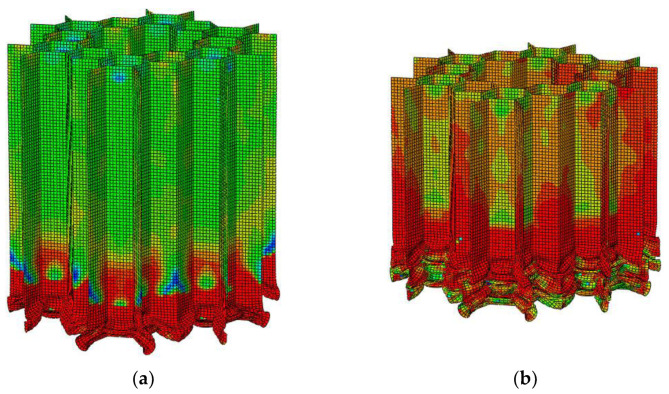
The deformation process of out-of-plane impact: (**a**) 20 mm; (**b**) 40 mm; (**c**) 60 mm; (**d**) 74 mm.

**Figure 12 materials-16-07121-f012:**
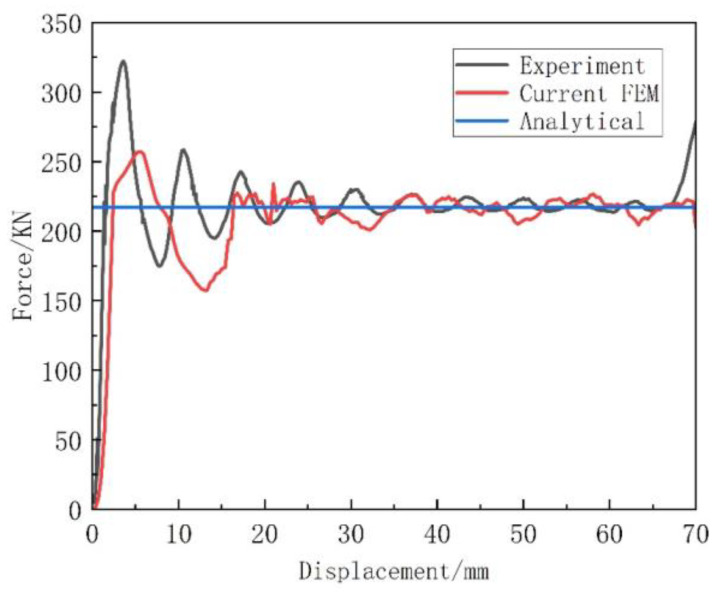
Comparison with Fang’s results.

**Figure 13 materials-16-07121-f013:**
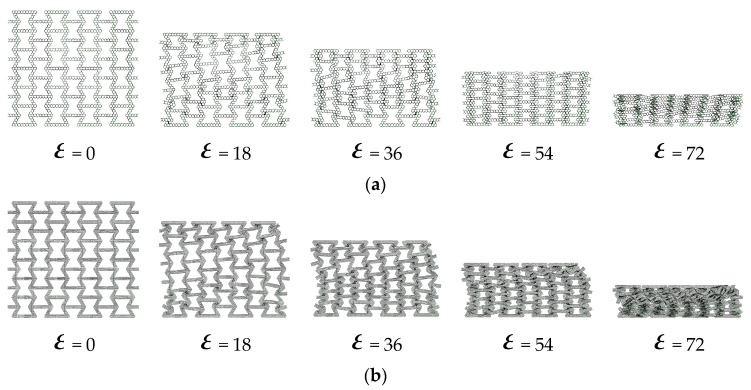
The deformation process of in-plane impact: (**a**) HEX-1; (**b**) HEX-2; (**c**) HEX-3.

**Figure 14 materials-16-07121-f014:**
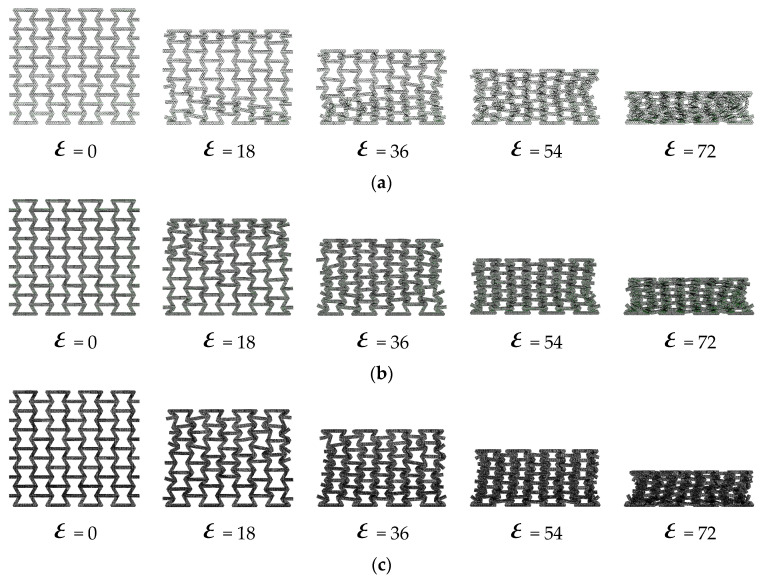
The deformation process of in-plane impact: (**a**) TRI-1; (**b**) TRI-2; (**c**) TRI-3.

**Figure 15 materials-16-07121-f015:**
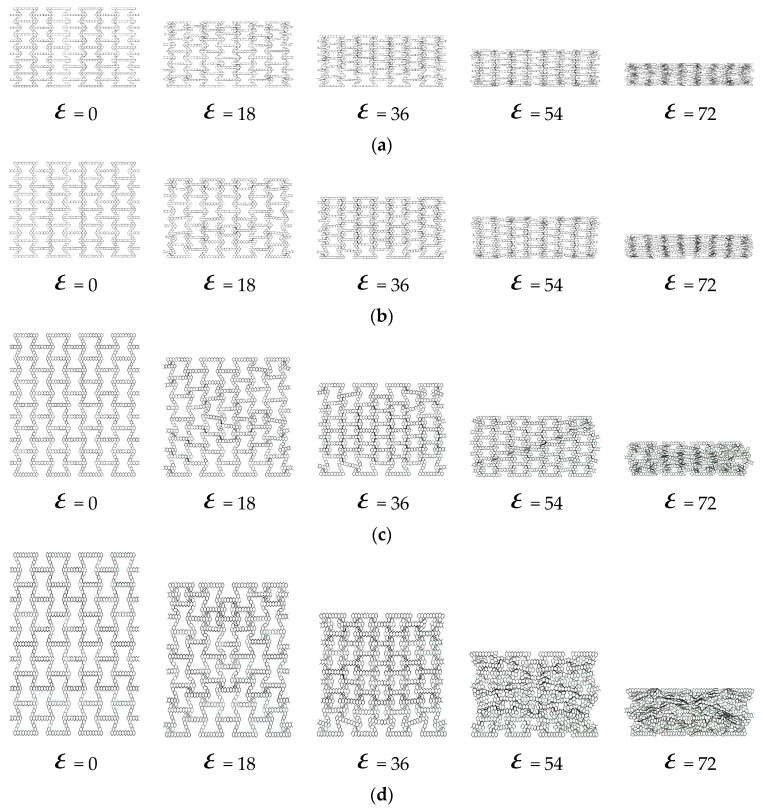
The deformation process of in-plane impact: (**a**) HEX-100°; (**b**) HEX-110°; (**c**) HEX-130°; (**d**) HEX-140.

**Figure 16 materials-16-07121-f016:**
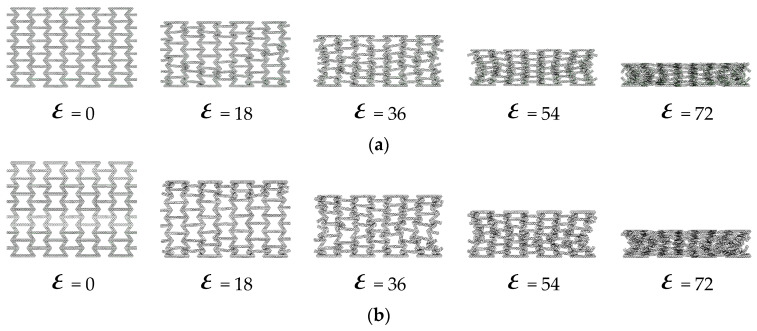
The deformation process of in-plane impact: (**a**) TRI-50°; (**b**) TRI-55°; (**c**) TRI-65°; (**d**) TRI-70°.

**Figure 17 materials-16-07121-f017:**
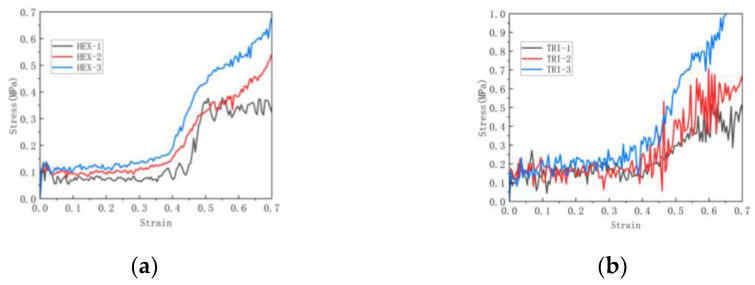
Stress–strain curve of in-plane impact: (**a**) Variable number hexagonal substructure; (**b**) Variable number triangular substructure.

**Figure 18 materials-16-07121-f018:**
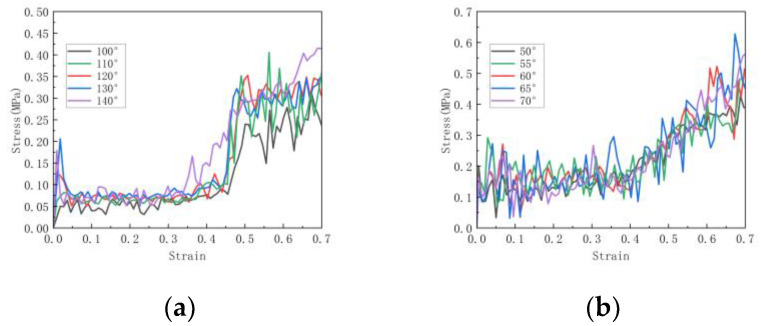
Stress–strain curve of in-plane impact: (**a**) Variable angle hexagonal substructure; (**b**) Variable angle triangular substructure.

**Figure 19 materials-16-07121-f019:**
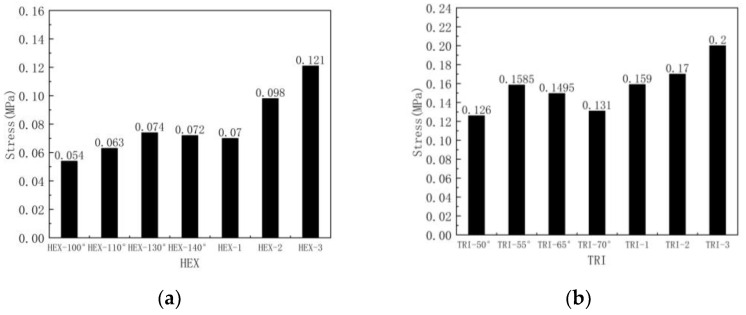
Platform stress of in-plane impact: (**a**) hexagonal substructure; (**b**) triangular substructure.

**Figure 20 materials-16-07121-f020:**
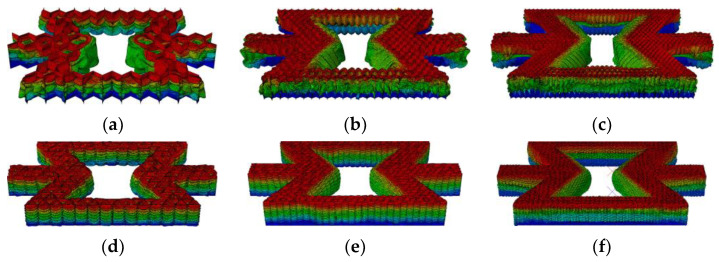
Deformation process of out-of-plane impact: (**a**) HEX-1; (**b**) HEX-2; (**c**) HEX-3; (**d**) TRI-1; (**e**) TRI-2; (**f**) TRI-3.

**Figure 21 materials-16-07121-f021:**
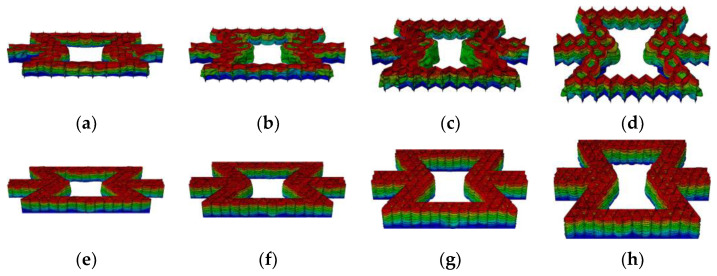
Deformation process of out-of-plane impact: (**a**) HEX-100°; (**b**) HEX-110°; (**c**) HEX-130°; (**d**) HEX-140°; (**e**) TRI-50°; (**f**) TRI-55°; (**g**) TRI-65°; (**h**) TRI-70°.

**Figure 22 materials-16-07121-f022:**
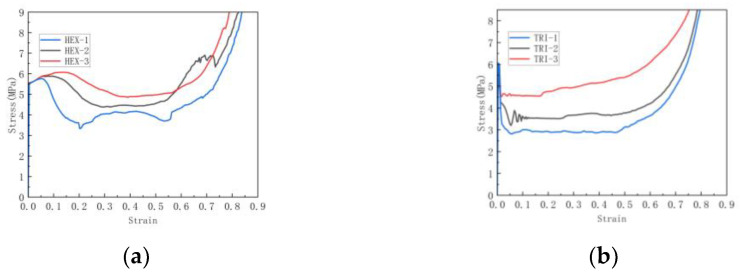
Stress–strain curve of out-of-plane impact: (**a**) Variable number hexagonal substructure; (**b**) Variable number triangular substructure.

**Figure 23 materials-16-07121-f023:**
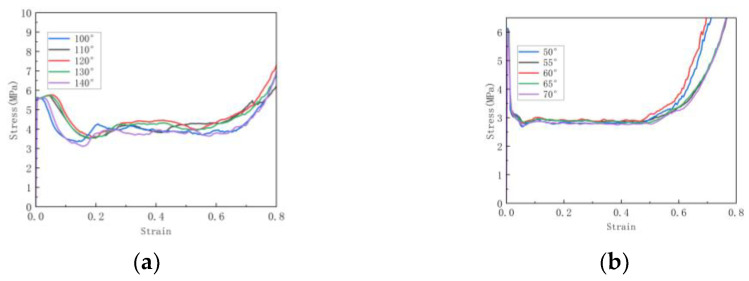
Stress–strain curve of out-of-plane impact: (**a**) Variable angle hexagonal substructure; (**b**) Variable angle triangular substructure.

**Figure 24 materials-16-07121-f024:**
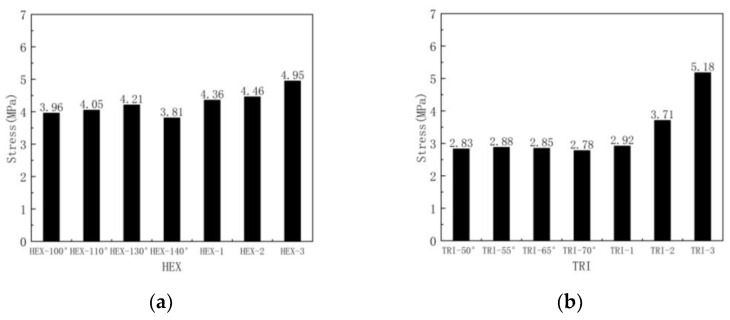
Platform stress of out-of-plane impact: (**a**) hexagonal substructure; (**b**) triangular substructure.

**Table 1 materials-16-07121-t001:** Variable substructure number and honeycomb parameters.

	θ	L (mm)	H (mm)
HEX-1	120°	12	1.07
HEX-2		4.8	0.48
HEX-3		3	0.31
TRI-1	60°	13.86	0.438
TRI-2		6.93	0.234
TRI-3		3.46	0.121

**Table 2 materials-16-07121-t002:** Variable substructure angle and honeycomb parameters.

	θ	L (mm)	H (mm)
HEX-1	100°	12	0.951
	110°	12	1.038
	130°	12	1.036
	140°	12	0.934
TRI-1	50°	15.76	0.4
	55°	15.04	0.437
	65°	12.26	0.429
	70°	10.28	0.396

## Data Availability

Data are contained within the article.

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
