# Peer review of "Impact Response of Re-Entrant Hierarchical Honeycomb"

_materials, 2023, doi:10.3390/ma16227121_

Round 1

Reviewer 1 Report

Comments and Suggestions for Authors

Section wise comments are as follows:

1. Introduction

The authors provide a well detailed introduction to honeycomb panels and the graded honeycomb structure being analyzed in this work. Sufficient background  work has been cited and the objectives of the paper laid out very well.

2. Hierarchical  Honeycomb

Please correct font size, style and spacing. It is not uniform throughout the section.

Mathematical models have been adequately described and overall analysis methodology well explained.

3. Numerical analysis

Please correct font size, style and spacing. It is not uniform throughout the section

Sections 4 and 5 are well described, no suggestions for further improvement.

Author Response

Dear reviewer, the font size, style and spacing you mentioned have been corrected.

Reviewer 2 Report

Comments and Suggestions for Authors

By replacing the cell wall of the concave honeycomb with hexagonal and triangular substructures with different angles and numbers of substructures, a honeycomb structure is proposed. The compressive strength was theoretically analyzed. I have the following comments: 1. What is the resistance to buckling, especially at high slice heights? 2. How the patch will deform when the compressive force acts obliquely 3. Where the proposed solution is particularly useful, I mean industrial applications. 4. Were any tests performed on a real object? Comments on the Quality of English Language The language is understandable to me. I don't feel competent to assess the quality of the language.

Author Response

Hello, Dear reviewer, I apologize for the trouble I have caused you due to my English level.   Here is my response to your suggestion.
1.    What is the resistance to buckling, especially at high slice heights?
It can be seen from Reference 28 that for honeycombs, buckling does not affect the platform stress when the honeycomb is subjected to out-of-plane impact.  In this paper, the initial defects are introduced when compared with Reference 28, and the results are basically consistent with the literature results.     The effect of buckling on in-plane impact is small.   For in-plane impact, the out-of-plane thickness of this paper is 2mm, but in actual production, the thickness of the honeycomb is relatively large.     Therefore, in the calculation process, 2mm is taken to shorten the workload and limit the displacement in the z direction when the in-plane impact occurs.
2.    How the patch will deform when the compressive force acts obliquely
When the impact is tilted, the deformation of the honeycomb is irregular, which makes it impossible to derive the specific platform stress of the honeycomb by formula.     At present, there are few studies on the tilt impact of the honeycomb, so it is impossible to verify the reliability of the model when the tilt impact occurs with my ability.     Therefore, this article does not study the tilt impact of the honeycomb, for which I am deeply sorry.
3.    Where the proposed solution is particularly useful, I mean industrial applications.
In the conclusion stage, the contribution of this paper to honeycomb research is added.
4.    Were any tests performed on a real object?
Due to the limited conditions and the complexity of the model, this paper cannot be compared through experiments, which I am sorry.     However, this paper verifies the reliability of the model through experiments with others.

This article has improved the language level.

Reviewer 3 Report

Comments and Suggestions for Authors

1- How the deformation process of out of plane is simulated by the mathematical equations.

2- It should be explained that how the study can enhance the designing process of honeycombs.

Comments on the Quality of English Language

1- How the deformation process of out of plane is simulated by the mathematical equations.

2- It should be explained that how the study can enhance the designing process of honeycombs.

Author Response

Dear reviewer
1. How the deformation process of out of plane is simulated by the mathematical equations.
In the verification stage of the article, the platform stress of in-plane and out-of-plane impact is added respectively.
2. It should be explained that how the study can enhance the designing process of honeycombs.
In the conclusion stage, the contribution of this paper to honeycomb research is added.

I apologize again for the trouble caused by the lack of my English level, and have further modified the language of this article.  I hope to reduce the trouble to reviewers and editorial board members.